# The Squeaky Yeast Gets Greased: The Roles of Host Lipids in the Clearance of Pathogenic Fungi

**DOI:** 10.3390/jof6010019

**Published:** 2020-01-31

**Authors:** Gaelen Guzman, Patrick Niekamp, Fikadu Geta Tafesse

**Affiliations:** 1Department of Molecular Microbiology and Immunology, Oregon Health & Science University, Portland, OR 97239, USA; guzmanga@ohsu.edu (G.G.); Patrick.Niekamp@Biologie.Uni-Osnabrueck.DE (P.N.); 2Biology & Chemistry Department, University of Osnabrück, Fachbereich Biologie/Chemie, Barbarastrasse 13, 49076 Osnabrück, Germany

**Keywords:** phagocytosis, phagolysosomal maturation, fungal pathogens, sphingolipids, phosphoniositides

## Abstract

Fungal infections remain a global health threat with high morbidity and mortality. The human immune system must, therefore, perpetually defend against invasive fungal infections. Phagocytosis is critical for the clearance of fungal pathogens, as this cellular process allows select immune cells to internalize and destroy invading fungal cells. While much is known about the protein players that enable phagocytosis, the various roles that lipids play during this fundamental innate immune process are still being illuminated. In this review, we describe recent discoveries that shed new light on the mechanisms by which host lipids enable the phagocytic uptake and clearance of fungal pathogens.

## 1. Introduction

Humans must maintain a balletic coexistence with a variety of environmental fungi, such as *Candida albicans*, which are ubiquitous in the environment and common commensal members of the human microbiome but are primed for opportunistic pathogenicity. Alongside *Candida*, the genera *Aspergillus* and *Cryptococcus* continuously challenge the human immune system. Resultantly, a host of innate immune mechanisms act in concert to sense and combat fungal infections. Phagocytosis is a central mechanism in the clearance of pathogens in which designated innate immune cells use receptors to recognize pathogen-associated molecular patterns displayed on fungal cell walls. Once internalized, the pathogen is destroyed in a microbicidal vacuole called the phagolysosome. A large body of literature delineates the duality of the fungal lifestyle, as well as the interactions between fungal pathogens and the human immune system [1,2]. However, the roles of host lipids during the resolution of fungal infections remain poorly understood. Here, we summarize and discuss the state of the current knowledge about the interplay between the host lipids and fungi during infection. We specifically focus on recent findings that show that sphingolipid and phosphoinositide metabolism is critical in enabling the phagocytosis and the subsequent phagolysosomal degradation of fungal pathogens. This summary presents several manners in which lipids are known to play essential roles during phagocytosis.

## 2. The Offense

Three genera are responsible for the majority of fungal infections: *Candida*, *Aspergillus*, and *Cryptococcus*. Species of these genera are ubiquitous in the environment and are common constituents of the human microbiome. Primarily, they exhibit their opportunistic pathogenicity in immunocompromised individuals [2]. Candida species such as *C. albicans* maintain two morphologies, a yeast form associated with commensalism and a hyphal form associated with virulence. Candida species are most-commonly associated with vulvovaginal candidiasis but systemic infections are common in hospitalized patients, with candidemia ranked at the 10th most common bloodstream infection [1,3]. Among the Aspergilli, *A. fumigatus* is the most common species to cause infection. *A. fumigatus* is an airborne pathogen that has few pathological effects on immunocompetent hosts, but can rapidly progress to a fatal invasive pulmonary aspergillosis in individuals undergoing immunosuppressive therapies. Cryptococci are also highly prevalent in the environment, and species such as *C. neoformans* can initiate a pulmonary infection, which progresses to a lethal meningitis in immunocompromised hosts.

While antifungal therapies have been effective in the past, resistance is increasingly emerging. For extensive appraisal on the changing epidemiology of invasive fungal infections, we direct readers to the 2017 reviews by Enoch et al.ia and Benedict et al.ia [4,5]. 

## 3. The Defense

An invasive fungal infection begins with penetration of a barrier surface, such as the lung, intestinal, or vaginal epithelium, followed by systemic dissemination. Consequently, the innate immune defenses are layered to catch incipient infections first at these barrier sites and then at centralized sites such as the liver and spleen. The primary cell types responsible for clearing a fungal infection are phagocytes such as macrophages, dendritic cells, and neutrophils. 

Immune surveillance is conducted at epithelial surfaces by a defense network of resident macrophages and dendritic cells. These cells express an array of germline-encoded pattern recognition receptors (PRRs), which recognize a variety of pathogen-associated molecular patterns (PAMPs). Table 1 depicts the list of the main PRRs known to recognize fungus-specific PAMPs. Upon ligand engagement, these receptors are broadly responsible for initiating pro-inflammatory cytokine responses and phagocytic uptake of the invading fungus. Following inflammatory activation by macrophages and dendritic cells, neutrophils are rapidly recruited to the site of barrier breach. These cells are the so-called “first responders” of the innate immune system, and are significantly primed for antimicrobial activity. 

After the killing of the fungal cell, the phagocyte finds itself with a surfeit of uniquely fungal molecular components that may be used to train the adaptive immune repertoire to prime the host for defense against future infection. To this end, the phagocyte loads phagolysosomal detritus onto antigen-presentation complexes in order to direct clonal selection of antigen-specific T cells. The most well-known of these complexes are the class I and II major histocompatibility complexes (MHC-I and –II, respectively), which are loaded with peptides. However, a parallel series of antigen presentation complexes, the CD1 family, allows for the presentation of lipid structures in order to drive clonal selection of natural killer T cells (NKT) [22,23]. Kawakami et al. showed in 2001 that these NKT cells promote the development of an antigen-specific Th1 cell response and enhance resistance to *C. neoformans* reinfection [24]. Of particular note, Albacker et al. were the first to identify a fungal glycolipid which directly activates NKT cells via presentation on CD1; this lipid species was the glycosphingolipid asperamide B, isolated from *A. fumigatus* [25]. Thus, the presentation of fungal lipids to the adaptive immune compartment is an important component of host defense against future exposure to pathogenic fungi.

Seminal works by Bird et al. in 1970, Baine et al. in 1974, and Sawyer et al. in 1976 show that specialized macrophages in the liver, spleen, and lymph system are essential for clearing fungal pathogens from the bloodstream and interstitium in simulated systemic infections [26,27,28]. Thus, multiple layers of defense protect an immunocompetent host from fungal infections, both localized and systemic. Despite their distinct roles, macrophages, dendritic cells, and neutrophils all employ phagocytosis as a central mechanism for clearing fungal pathogens. As will be discussed below, lipids play essential roles in the uptake and clearance of pathogenic fungi.

## 4. Phagocytosis: The Weapon of Choice

Phagocytosis is a form of receptor-mediated endocytosis characterized by its dependency on actin cytoskeletal rearrangement, and its ability to mediate the internalization of particles larger than 0.5 µm in diameter [29]. Following internalization, the ingested particle or fungal cell encapsulated in the phagosome undergoes sequential maturation stages until it terminally fuses with the lysosome to form the phagolysosome, and is therein destroyed. The process of uptake involves highly orchestrated signaling interactions between GTPases, tyrosine and serine/threonine kinases, phospholipases, cytoskeletal components, and lipid species such as phosphoinositides [29,30]. While phagocytosis has been studied extensively over the last century, significant knowledge regarding the roles of lipids during this essential immunological process has been gleaned only recently. Intuitively, dramatic membrane reorganization must occur for the internalization of particles >0.5 µm in diameter, but techniques and tools which visualize these membrane dynamics are relatively recent advents for biologists [31]. 

Macrophages and dendritic cells constitutively sample and probe their environment using actin-driven protrusions. If a phagocytic receptor engages its cognate ligand on a particle, it tethers that particle to the cell and more receptors engage to form a signaling platform known as the “phagocytic synapse” [32]. This signaling platform closely resembles the specialized membrane microdomains known as lipid rafts. Proteomics studies of the phagosome revealed several proteins associated with isolated detergent-resistant membrane domains (one functional definition of a lipid raft), and the signaling capacity of the fungal receptor Dectin-1 was shown to be sensitive to treatment with the lipid raft-disrupting methyl-β-cyclodextrin [33,34,35]. As depicted in Figure 1A, both sphingomyelin and cholesterol are essential lipid components of lipid rafts, and are believed to be enriched to the phagocytic synapse. Of particular note, fluorescent sphingomyelin analogs reported by Kinoshita et al. in 2017 allow for the visualization of sphingomyelin in raft domains, and were used to demonstrate that sphingomyelin is recruited to the phagocytic synapse of IgG opsonized particles [36,37]. 

Similarly, we and others demonstrate that sphingolipid biosynthesis is essential for phagocytic uptake and clearance of *C. albicans* and other microbial pathogens [38,39,40,41]. In particular, we demonstrated evidence that sphingolipid biosynthesis is essential for formation of the phagocytic synapse upon contact with Zymosan A, a model particle for fungal cells composed of β-1,3-glucans. In sphingolipid-deficient cells, we observe that the phagocytic receptor Dectin-1 and the anti-phagocytic phosphatase CD45 remain colocalized at the contact site [39]. Thus, membrane order and composition at the phagosomal synapse have significant influences on the clearance of fungal pathogens. 

Signaling lipids such as phosphatidylinositides are known to directly participate in the phagocytic signal cascade. Figure 1B depicts the interconversion of phosphatidylinositide species during phagocytosis. Briefly, phagocytic receptor aggregation at the phagocytic synapse induces a potent phosphorylation cascade which begins with the activation of tyrosine kinases, followed by the extension of actin-rich membrane protrusions, and culminates in actin disassembly and internalization of the particle [10,33,42]. One such tyrosine kinase, Syk, activates several enzymes that modify phosphatidylinositol-4,5-*bis*phosphate (PIP_2_), such as phosphatidylinositol-3 kinase (PI3K) which converts PIP_2_ to phosphatidylinositol-3,4,5-*tris*phosphate (PIP_3_), and phospholipase Cγ which produces diacylglycerol and soluble inositol(1,4,5)-*tris*phosphate (IP_3_) [43]. PIP_3_ serves as a membrane anchor and recruitment signal for proteins essential for phagocytic uptake [44]. For example, both the actin nucleation promoter Scar/WAVE and MyosinX contain so-called Plekstrin Homology (PH) domains with high affinity for PIP_3_, meaning that the Syk-dependent activation of PI3K directly results in the recruitment of these cytoskeletal elements to the membrane [45]. Similarly, soluble IP_3_ serves as a secondary messenger that induces calcium flux from the endoplasmic reticulum. Evidence suggests that the resulting cytosolic burst of calcium is essential for the maturation of the phagosome and the lysosome, and PIP_2_ metabolism is thereby central to the antimicrobial activity of phagocytes [46,47]. 

Lipid metabolism also plays essential roles in modulating the fluidity and curvature of the plasma membrane at the phagocytic synapse. Early evidence by Hauck et al. shows that the activity of acid sphingomyelinase (ASM) is essential for uptake of the bacterial pathogen *Neisseria gonorrheae* [48]. A potential explanation for the role of sphingomyelin-derived ceramide during phagocytosis may lie in the well-known role ceramide has in inducing negative membrane curvature and vesicular budding [49,50]. Sphingomyelin has a characteristic cylindrical shape and preferentially maintains rigid, planar membrane structures such as raft-like domains, whereas ceramide’s small headgroup induces invaginating membrane structures [50]. Similarly, the conversion of phosphatidylcholine to phosphatidic acid by phospholipase D during phagocytosis has been suggested to directly induce negative membrane curvature and thereby enhance the internalization of pathogens such as fungi [51,52]. Alternatively, it has been suggested that, at high localized concentrations, ceramide can assemble in hexagonal phase II structures in which the membrane bilayer is disrupted and becomes highly flexible. These two mechanisms may together explain the pro-fusion effect of ceramide production during phagosomal maturation [50].

## 5. The Phagolysosome: The Final Stage for Eliminating Fungal Pathogens

Phagocytic uptake alone is not sufficient for an innate immune cell to destroy an invasive fungal pathogen; this task is consigned to the lysosome. Following uptake, the fungus-containing phagosome is trafficked along the endocytic pathway, gradually becoming more acidic before fusing with the lysosome [30,53,54]. The resulting phagolysosome is a hostile environment that serves to degrade the phagosome contents to base molecular components. Few organisms are capable of surviving the extremely acidic pH, protease and nuclease activity, reactive oxygen and nitrogen species, pore-forming peptides, and metal ion depletion within the lysosome [55]. Significant evidence demonstrates that lipid metabolism plays essential roles throughout phagosomal maturation [50,56,57]. In particular, several lysosomal storage disorders are associated with high susceptibility to fungal infections. 

The maturation of the phagosome requires a number of membrane fission and fusion events which are significantly dependent on the activity of lipid metabolizing enzymes. For example, extensive reviews describe the role of phosphoinositides as a coordinator of phagosome maturation [30,45,58,59]. The small GTPase Rab5 is quickly recruited to the newly-formed phagosome, and directly activates the class III phosphatidylinositol 3-kinase human vacuolar protein-sorting 34 (hvPS34). This kinase converts phosphatidylinositol to phosphatidylinositol-3-phosphate, which serves as a membrane anchor for proteins containing either PX (phagocytic oxidase) domains or FYVE (Fab1, YOTB, Vac 1, and EEA1) domains. One such FYVE-domain-containing protein is the early endosome antigen 1 (EEA1), which serves as a bridge to the early endosome by interacting with syntaxin 13, a member of the SNARE family of membrane fusors. Because phosphatidylinositol-3-phosphate production is essential for coordinating the interaction between EEA1 on the newly-formed phagosome and syntaxin 13 on the early endosome, this phosphoinositide plays a direct role in the maturation of the phagosome [30,45,58,59]. Phosphoinositides are not the only lipid family required for proper phagosome maturation. As depicted in Figure 1C, the regulated conversion of sphingomyelin to ceramide by ASM within the maturing phagosome is essential for its propensity for fusion between the early and late endosomes and the lysosome [50]. 

Lipid metabolism continues to be essential for the neutralization of phagocytosed pathogens even after fusion with the lysosome. For example, it is well known that lysosomal ceramide interacts with and activates the aspartyl protease Cathepsin D [60]. The Ramakrishnan group has shown that lysosomal ceramide produced by ASM is required for the activation of Cathepsin D in a mycobacteria-induced necrosis circuit [61,62]. This role for the conversion of sphingomyelin to ceramide is depicted in Figure 1D. Lipidomic analysis of the composition of the maturing phagosome showed that the total levels of sphingomyelin decrease as the phagosome travels along the endocytic compartment, and there is sparingly little sphingomyelin by the time the phagosome merges with the lysosome [63]. Concordantly, patients with disorders associated with deficiencies of ASM function, such as Niemann Pick Disease of types A and B, are highly susceptible to recurrent infection by bacterial and fungal pathogens, as these patients are unable to efficiently eliminate pathogens after internalization [51,57]. 

In a related example, the genetic disorder cystic fibrosis culminates in an overabundance of ceramide in the lysosome due to improper regulation of ASM and acid ceramidase activities [57]. These individuals have an increased risk of chronic or recurrent infections by bacterial and fungal pathogens. It was shown that treatment with the ASM inhibitor amitriptyline normalized the balance of ceramide in the lysosome and prevented the chronic pulmonary inflammation and reversed susceptibility to *P. aeruginosa* in mouse models [56]. Thus, genetic disorders with opposing effects on ASM activity may result in the convergent phenotype of increased susceptibility to infection, highlighting the tight regulation on lipid metabolism required for proper clearance of infection. 

## 6. Conclusions and Perspectives

Invasive fungal infections by Candida, Aspergillus, and Cryptococcus species are a developing public health concern. While these opportunistic pathogens largely act as benign environmental fungi or commensals in immunocompetent humans, they pose a significant risk to immunocompromised individuals. As modern medicine progresses, the prevalence of patients on immunosuppressive therapies such as organ transplant recipients and those with latent retroviral infections is increasing. As such, invasive fungal infections are poised to become a significant health crisis. The vignettes presented in this review are intended to highlight several of the well-known manners in which lipids are critical for the uptake and clearance of fungal pathogens. Recent years have shown that sphingolipids and phosphoinositides serve both as essential regulators of membrane fluidity and structure and potent signaling molecules. 

Many open questions remain regarding the mechanisms by which pathogenic fungi may usurp host lipid metabolism pathways to effectively establish infection. Further studies are required to define the exact roles of myriad host lipid species in fungal pathogenesis and host immunity. Recent work has demonstrated that lipids found on the surface of the bacteria *Mycobacterium tuberculosis* may directly influence the fluidity of the host phagocyte’s plasma membrane during uptake [64], but no work has yet been reported that would suggest that fungal lipids may play a similar direct role in influencing host membrane architecture. Similarly, many non-fungal pathogens are known to influence host lipid metabolism networks [65,66,67], but little is known whether fungi similarly influence host lipid metabolism. 

It is likely that many new findings will come to light in the coming years. With the burgeoning of the biologists’ toolset, there is growing recognition that lipids are not merely the inert building blocks of membranes but rather active interaction partners and signaling molecules. A greater understanding of innate immunological mechanisms might provide insights into how immune cells utilize effector molecules such as lipids to selectively destroy invading fungi, which may, thus, reveal novel mechanisms for antifungal therapies. 

## Figures and Tables

**Figure 1 jof-06-00019-f001:**
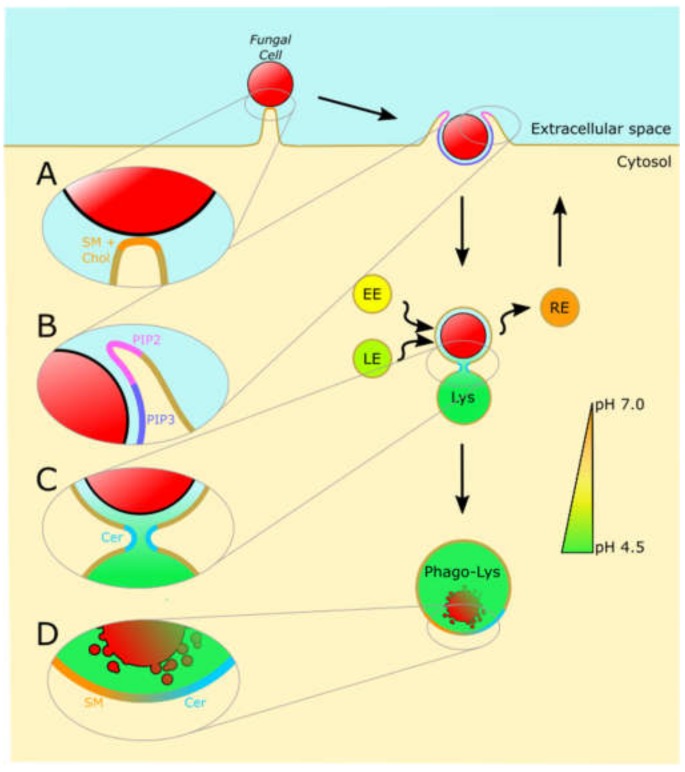
Host lipids are critical for the uptake and destruction of fungal pathogens. A simplified depiction of several stages at which lipids are known to be essential for the clearance of fungi by phagocytic immune cells. (**A**) Upon the initial recognition of a fungal cell, pathogen recognition receptors cluster in a microdomain reminiscent of a lipid raft. This domain is enriched in sphingomyelin (SM) and cholesterol (Chol), and serves to form a signaling platform from which the phagocytic process is coordinated. (**B**) The turnover of phosphoinositide species is essential to the extension of pseudopodia around a particle. Emanating from the signaling platform at the center of the phagocytic synapse is a leading wave of PIP_2_, followed by a trailing wave of PIP_3_. High levels of PIP_2_ initiate filamentous actin polymerization and thereby initiate the actin-driven protrusion which extends the pseudopodia. The conversion of PIP_2_ to other lipid species such as PIP_3_ is essential for actin disassembly and recycling. (**C**) After internalization, lipids such as ceramide (Cer) are essential for the fusion and fission of the phagosome with the compartments of the endosomal pathway, such as the early endosome (EE), late endosome (LE), recycling endosome (RE), and lysosome (Lys). (**D**) After fusion with the lysosome, the degradation of sphingomyelin to ceramide is essential for the activation of lysosomal components such as cathepsins, which cooperate to destroy the fungal cell.

**Table 1 jof-06-00019-t001:** Fungus-specific pattern recognition receptors and their cognate molecular patterns.

PRR	PAMP	References
Mannose receptor	Mannan, mannoproteins	[6,7,8,9]
Dectin-1	β-1,3 glucan	[9,10,11,12,13]
Dectin-2	α-mannans	[2,13]
Mincle	α-mannose	[9,13,14,15]
DC-SIGN	Galactomannans	[9,13,16]
Galectin-3	β-1,2 mannosides	[9,17]
Complement receptor 3	C3b, β-glucans	[9,18]
Toll-like receptor 2	Chitin and other polysaccharides	[9,19,20]
Toll-like receptor 4	O-linked mannosyl residues	[21]

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
