# Peer review of "The Squeaky Yeast Gets Greased: The Roles of Host Lipids in the Clearance of Pathogenic Fungi"

_jof, 2020, doi:10.3390/jof6010019_

Round 1

Reviewer 1 Report

The review article by Guzman, Niekamp, and Tafesse provides a nice overview of the current understanding of lipids in phagocytosis related to fungal infection. The paper is well-written and clear. A few minor items require attention:

Figure 1 should be referenced in the body (text) of the review. In the legend for Figure 1, the usage of “invading” (line 146) may be confusing. Do the authors simply mean a fungal cell that has ‘infected’ the host? Or do the authors mean that fungal cell is going to invade the phagocytic cell? If it is the latter, then an invading fungal cell may not necessarily be destroyed as some pathogens may invade (i.e. enter) cells including macrophages to replicate or evade the immune response. The authors should consider removing “invading”. Although Table 1 is a nice summary, its appropriateness in this review is tenuous. Perhaps a short table summarizing the highlight papers that examine the role of lipids in phagocytosis would be more appropriate. Lines 25-26, “immune” is redundant. The authors could simply state: “…designated innate immune cells use receptors to….” The last two sentences (lines 35-38) of the Introduction seem more appropriate to the “Conclusion and Perspectives” section. The authors may consider moving them. Much of the review discusses the role of lipids during phagocytosis in general. Is there aspect to lipids in this process that is unique to fungal cells? Perhaps they can comment on how this is an unexplored area of biology.

Author Response

Point 1: Figure 1 should be referenced in the body (text) of the review.

We have now included references throughout the body to each of the relevant sections of the figure to direct the reader.

Point 2: In the legend for Figure 1, the usage of “invading” (line 146) may be confusing. Do the authors simply mean a fungal cell that has ‘infected’ the host? Or do the authors mean that fungal cell is going to invade the phagocytic cell? If it is the latter, then an invading fungal cell may not necessarily be destroyed as some pathogens may invade (i.e. enter) cells including macrophages to replicate or evade the immune response. The authors should consider removing “invading”.

Thank you for this comment - we intended the word "invading" to denote a fungal cell infecting the host, not the phagocytic cell. The word "invading" has been removed for clarity.

Point 3: Although Table 1 is a nice summary, its appropriateness in this review is tenuous. Perhaps a short table summarizing the highlight papers that examine the role of lipids in phagocytosis would be more appropriate.

We have discussed this at length, and feel that there is value in maintaining this table - the receptors/ligands listed are considered the most critical in the identification of fungal pathogens, and are essential for initiating the antifungal activity of the phagocyte by interacting with different lipids (Section 3 and 4).   

Point 4: Lines 25-26, “immune” is redundant. The authors could simply state: “…designated innate immune cells use receptors to….”

We have taken your suggestion and removed the redundant wording.

Point 5: The last two sentences (lines 35-38) of the Introduction seem more appropriate to the “Conclusion and Perspectives” section. The authors may consider moving them.

These sentences have been moved to the conclusions section as advised, thank you.

Point 6: Much of the review discusses the role of lipids during phagocytosis in general. Is there aspect to lipids in this process that is unique to fungal cells? Perhaps they can comment on how this is an unexplored area of biology. 

To expand on the role of lipids in a fungus-specific context, we have added a paragraph (Section 3) describing the presentation of fungal lipid structures to the adaptive immune system. The most well understood example is the A. fumigatus glycosphingolipid Aperamide B, which can directly activate a localized antimicrobial immune state when presented on CD1 by phagocytes. We have also edited the conclusion section to highlight several outstanding questions regarding the unexplored area of fungal lipids and host lipid metabolism during fungal infection. We hope that this address the reviewer's concerns.

Reviewer 2 Report

The authors Guzman, Niekamp, and Tafesse, have nicely reviewed the roles of host lipids during the immune response to fungi. In particular they have focused on the involvement of lipids during phagocytosis of yeasts. Thereby they review the action of sphingomyelins in the phagocytic synapse of dendritic cells and macrophages, the role of phosphatidylinositols PIP3 and PIP2 and phospholipases during phagocytic uptake, and finally the involvement of ceramides during phagosomal maturation and formation of the phagolysosome. What I found a little difficult to discern from the manuscript was the question which lipids are relevant for the phagocytosis of fungi versus those of other particles or microorganisms. Are there any fungus specific lipid interactions? Are there any interactions of host lipids with the fungus expressed ergosterol or its metabolites? The therapeutic targets mentioned in the conclusion regard the fungal cell wall components and its lipids and seem to somewhat divert from the original focus on host lipids.

Minor point: Figure 1 should be reference to in the text. For example the paragraph that introduces signaling lipids in the phagocytic signal cascade would be a good position to reference to Figure 1.

Author Response

Point 1: What I found a little difficult to discern from the manuscript was the question which lipids are relevant for the phagocytosis of fungi versus those of other particles or microorganisms. Are there any fungus specific lipid interactions? Are there any interactions of host lipids with the fungus expressed ergosterol or its metabolites?

Thank you for your comment. To expand on the role of lipids in a fungus-specific context, we have added a paragraph (Section 3) describing the presentation of fungal lipid structures to the adaptive immune system. The most well understood example is the A. fumigatus glycosphingolipid Aperamide B, which can directly activate a localized antimicrobial immune state when presented on CD1 by phagocytes. We have also edited the conclusion section to highlight several outstanding questions regarding the unexplored area of fungal lipids and host lipid metabolism during fungal infection. We hope that this address the reviewer's concerns.

Point 2: The therapeutic targets mentioned in the conclusion regard the fungal cell wall components and its lipids and seem to somewhat divert from the original focus on host lipids.

Thank you, we fully agree with the reviewer's comment, and have removed this paragraph from the conclusion. To expand on point 1, we have replaced the paragraph with one which highlights open questions.

Point 3: Figure 1 should be reference to in the text. For example the paragraph that introduces signaling lipids in the phagocytic signal cascade would be a good position to reference to Figure 1.

Thank you for this comment, we have incorporated references to the figure throughout the body of the text in order to direct the reader to each relevant subsection of the figure.